# Natural Variation in Diauxic Shift between Patagonian *Saccharomyces eubayanus* Strains

Jennifer Molinet,[a,b] Juan I. Eizaguirre,[c] Pablo Quintrel,[a,b] Nicolás Bellora,[d*] Carlos A. Villarroel,[a,e,f] Pablo Villarreal,[a,b] José Benavides-Parra,[b] Roberto F. Nespolo,[a,g,h,i] Diego Libkind,[c] Francisco A. Cubillos[a,b,g]

[a]ANID-Millennium Science Initiative-Millennium Institute for Integrative Biology (iBio), Santiago, Chile

[b]Departamento de Biología, Facultad de Química y Biología, Universidad de Santiago de Chile, Santiago, Chile

[c]Instituto Andino Patagónico de Tecnologías Biológicas y Geoambientales (IPATEC), Universidad Nacional del Comahue, CONICET, CRUB, San Carlos de Bariloche, Río Negro, Argentina

[d]Instituto de Tecnologías Nucleares para la Salud (INTECNUS), Consejo Nacional de Investigaciones Científicas y Técnicas (CONICET), San Carlos de Bariloche, Argentina

[e]Instituto de Ciencias Biológicas, Universidad de Talca, Talca, Chile

[f]Instituto de Investigación Interdisciplinaria (I3), Universidad de Talca, Talca, Chile

[g]ANID-Millennium Nucleus of Patagonian Limit of Life (LiLi), Valdivia, Chile

[h]Instituto de Ciencias Ambientales y Evolutivas, Facultad de Ciencias, Universidad Austral de Chile, Valdivia, Chile

[i]Center of Applied Ecology and Sustainability (CAPES), Santiago, Chile

Jennifer Molinet and Juan I. Eizaguirre contributed equally to this article. The authors order was decided based on who wrote the paper.

**ABSTRACT** The study of natural variation can untap novel alleles with immense value for biotechnological applications. *Saccharomyces eubayanus* Patagonian isolates exhibit differences in the diauxic shift between glucose and maltose, representing a suitable model to study their natural genetic variation for novel strains for brewing. However, little is known about the genetic variants and chromatin regulators responsible for these differences. Here, we show how genome-wide chromatin accessibility and gene expression differences underlie distinct diauxic shift profiles in *S. eubayanus*. We identified two strains with a rapid diauxic shift between glucose and maltose (CL467.1 and CBS12357) and one strain with a remarkably low fermentation efficiency and longer lag phase during diauxic shift (QC18). This is associated in the QC18 strain with lower transcriptional activity and chromatin accessibility of specific genes of maltose metabolism and higher expression levels of glucose transporters. These differences are governed by the HAP complex, which differentially regulates gene expression depending on the genetic background. We found in the QC18 strain a contrasting phenotype to those phenotypes described in *S. cerevisiae*, where *hap4Δ*, *hap5Δ*, and *cin5Δ* knockouts significantly improved the QC18 growth rate in the glucose-maltose shift. The most profound effects were found between *CIN5* allelic variants, suggesting that Cin5p could strongly activate a repressor of the diauxic shift in the QC18 strain but not necessarily in the other strains. The differences between strains could originate from the tree host from which the strains were obtained, which might determine the sugar source preference and the brewing potential of the strain.

**IMPORTANCE** The diauxic shift has been studied in budding yeast under laboratory conditions; however, few studies have addressed the diauxic shift between carbon sources under fermentative conditions. Here, we study the transcriptional and chromatin structure differences that explain the natural variation in fermentative capacity and efficiency during diauxic shift of natural isolates of *S. eubayanus*. Our results show how natural genetic variants in transcription factors impact sugar consumption preferences between strains. These variants have different effects depending on the genetic background, with a contrasting phenotype to those phenotypes previously described in *S. cerevisiae*. Our study shows how relatively simple genetic/molecular modifications/

Address correspondence to Francisco A. Cubillos, francisco.cubillos.r@usach.cl, or Diego Libkind, libkindfd@comahue-conicet-gob.ar.

*Present address: Nicolás Bellora, Instituto de Tecnologías Nucleares para la Salud (INTECNUS), Consejo Nacional de Investigaciones Científicas y Técnicas (CONICET), San Carlos de Bariloche, Argentina.

The authors declare no conflict of interest.

editing in the lab can facilitate the study of natural variations of microorganisms for the brewing industry.

**KEYWORDS** *Saccharomyces eubayanus*, wild strains, beer, RNA-seq, ATAC-seq, diauxic shift, HAP, wild, yeasts

The *Saccharomyces cerevisiae* domestication process represents a textbook example of the adaptation of microorganisms to anthropogenic settings (1). However, life in the wild for *Saccharomyces* species is still poorly understood (2). Since the recent discovery and identification of *Saccharomyces eubayanus* (3), one of the parents of the lager yeast hybrid, several studies developed biotechnological applications, together with research studies on its ecology, genetics, evolution, phylogeography, and natural history (4–10). *S. eubayanus* isolates have only been found in natural environments; however, the genetic material of *S. eubayanus* has been identified on multiple occasions in industrial hybrids, highlighting the existence of recurrent hybridization events between *Saccharomyces* species under the human-driven fermentative environment (3, 11–13). Fermentation at low temperatures carried out by the lager hybrid *Saccharomyces pastorianus* is undoubtedly the most important of these cases. It is clear then that *S. eubayanus*, analyzed in both natural and domesticated environments, represents an excellent experimental model for investigating physiological adaptation to both scenarios.

*S. pastorianus* is a classic example of hybrid vigor. It combines the cold tolerance of *S. eubayanus* together with the superior fermentation kinetics inherited from *S. cerevisiae*, both traits having synergistic effects on fitness, under the cold fermentative environment of European cellars since the middle ages (12, 14, 15). However, despite the importance of *S. pastorianus* in the brewing industry, the origin of this hybrid has not been entirely unraveled (12). This lack of information remains mostly because *S. eubayanus* has never been isolated in Europe, either in fermentation environments or in the wild (16). Paradoxically, this cryotolerant species has been extensively recovered in Argentina (3, 6), North America (17, 18), East Asia (19), New Zealand (20), and Chile (4). Although the evolutionary origin of this species is still a matter of debate, the large number of isolates, lineages, and the great genetic diversity found in the Andean region of Argentina and Chile support the hypothesis of a Patagonian origin for *S. eubayanus* (4, 5). This species is genetically structured into two main populations (PA and PB) and six subpopulations (HOL, PA-1, PA-2, PB-1, PB-2, and PB-3) (6, 17, 18), reflected in its biogeography (4, 5).

One of the most important parameters to evaluate in brewing fermentation is attenuation, that is, the ability of yeast to consume the sugars from the wort (21). A beer wort is mainly made up of glucose, maltose, maltotriose, and dextrins (22). While *S. pastorianus* can consume all of these sugars, *S. eubayanus* is capable of metabolizing just glucose and maltose but is unable to consume more complex sugars, such as maltotriose (23). Partial sugar consumption can result in sluggish fermentations that alter the sensorial properties of the final product (24, 25). Maltose is the most abundant sugar in the wort (approximately 60%) and the efficient consumption of this sugar is decisive in beer fermentation (26). In *S. cerevisiae*, three *MAL* loci control the utilization of maltose, where these genes encode a transcriptional activator (*MALx3*), a maltose permease (*MALx1*), and a maltase (*MALx2*). The numbers and identities of *MAL* loci are highly strain dependent, with up to five loci (*MAL1, 2, 3, 4,* and *6*) in haploid genomes (27). In the *S. eubayanus* CBS12357 type strain, four open reading frames share identity with the *S. cerevisiae MAL31* loci (*SeMALT1, SeMALT2, SeMALT3,* and *SeMALT4*), located in four subtelomeric regions where two of them contain the same organization described in *S. cerevisiae* (28, 29). However, the maltose consumption rate in *S. eubayanus* and other *Saccharomyces* species is slower compared to that of commercial lager strains, which are capable of rapidly fermenting all the sugars present in the wort (8, 30, 31). Furthermore, different *S. eubayanus* strains exhibit differences in their maltose

consumption rate and yield during fermentation (6, 8). The differences in fermentation rate originate from the phenotypic variation of the metabolic response to the presence of glucose in the medium. This sugar represses the expression of genes involved in the consumption of more complex sugars such as maltose, as well as genes involved in respiration, a phenomenon known as "glucose repression" (32). Glucose depletion activates a metabolic rewiring triggering the metabolism of other sugars, a process called diauxic shift, which may explain differences in the adaptation time between different *S. eubayanus* strains (33, 34). Then, a detailed molecular analysis of the diauxic shift in *S. eubayanus* strains is fundamental for the potential use of this wild yeast in the industry.

To identify the molecular origin of the variability in sugar consumption at low temperatures between natural *S. eubayanus* yeast strains, we studied the fermentation profiles of 19 strains of *S. eubayanus* that represent the different lineages distributed across Andean Patagonia. Given the great genetic diversity already documented in these lineages and the phenotypic changes observed in previous studies, we selected three strains from Patagonia to evaluate gene expression and chromatin structure differences in depth during the diauxic shift. This approach offered valuable information on the genetic variants underlying sugar consumption differences during fermentation, providing important insights into the ecology of the species and their potential application in the brewing industry.

## RESULTS

***S. eubayanus* Patagonian isolates exhibit differences in their fermentation profiles.** To determine the natural genetic variation in fermentative capacity under brewing conditions across Andean Patagonia *S. eubayanus* isolates, we initially selected 19 strains that are representative of the different geographic areas of Chile and Argentina (Fig. 1A; Table S1A in the supplemental material). After 14 days of beer fermentation, most strains exhibited similar kinetic profiles of $CO_2$ loss, except for the QC18 strain which exhibited the lowest fermentative capacity. In general, all strains showed a lower fermentative capacity than the commercial strain *S. pastorianus* W34/70. (Fig. 1B, $P < 0.05$, Student's *t* test, Table S2A and S2B). Sugar consumption and ethanol production differed across strains (Table S2C, S2D, and S2E). Interestingly, we observed lower levels of residual maltose and therefore incomplete fermentation in seven strains (CL204.3, CL211.3, CL215.1, CL218.1, CL444.4, CRUB2031, and CRUB 2108), where the QC18 strain exhibited the lowest consumption levels (ANOVA, $P < 0.0001$). However, we did not observe a correlation between residual maltose and $CO_2$ loss rate (Pearson correlation coefficient $-0.33$, $P = 0.14$, Fig. S1A) or maximum $CO_2$ loss (Pearson correlation coefficient $-0.14$, $P = 0.54$, Fig. S1A), likely because residual maltose was below 5 g/L. Instead, we found a significant correlation with ethanol production (Pearson correlation coefficient $-0.59$, $P = 0.005$, Fig. S1A). Notably, the strains differed in their amino acid consumption profiles at the end of the fermentation, demonstrating differences in their sugar and nitrogen consumption profiles (Table S2F, S2G, and S2H).

Principal component analysis (PCA) of six fermentation parameters (Fig. 1C) indicated that $CO_2$ loss rate and maximum $CO_2$ loss parameters correlate positively (Pearson correlation coefficient 0.56, $P = 7.3 \times 10^{-6}$, Fig. S1B), where the PC1 and PC2 components explain 42.4% and 24.3% of the observed variance, respectively. Interestingly, the individual factor map indicates no significant separation pattern according to geographical origin and/or phylogenetic group. In addition, we performed hierarchical clustering of the kinetic parameters, obtaining five main clusters (Fig. S1C). Again, we did not observe a significant pattern of divergence either by geographic origin or phylogenetic group.

To further explore potential differences between strains throughout the fermentation process, we selected three strains representative of the different phenotypes analyzed above. Of the 19 initial strains, the QC18 strain exhibited the worst kinetic parameters and was selected as a low-fermentation strain (LF). On the other hand, we selected strains CBS12357 and CL467.1 as representatives with higher fermentation capacities (HF). We evaluated the sugar consumption and ethanol production profiles of these

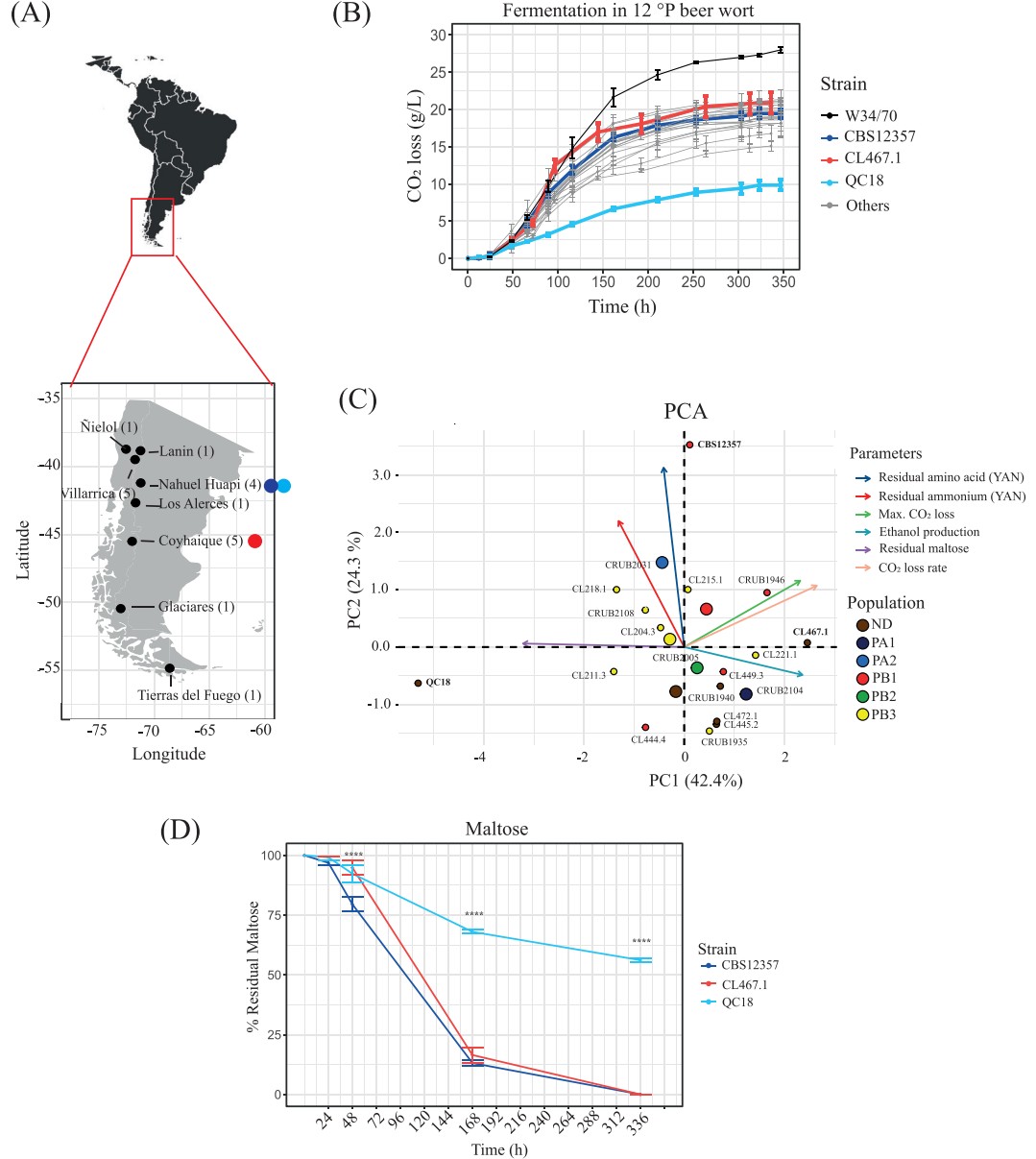

**FIG 1** Fermentation differences between Andean Patagonia *S. eubayanus* strains. (A) Map of Argentinian/Chilean Andean Patagonia together with the 8 localities from where the 19 strains were isolated. Red, blue, and light blue depict the three strains selected for the rest of this study. (B) $CO_2$ loss kinetics for 19 *S. eubayanus* strains and the *S. pastorianus* commercial strain W34/70. (C) Principal-component analysis (PCA) using fermentation parameters across the 19 strains, together with the distribution of individual strains. Arrows depict the different parameters. (D) Maltose consumption kinetics of CBS12357, CL467.1, and QC18 strains. Plotted values correspond to mean values of three independent replicates for each strain. ****, $P \leq 0.0001$, different levels of significance between QC18 and the other strains by $t$ test.

three strains at different time points during the fermentation process (24, 48, 168, and 336 h, Table S2C). Glucose and fructose were completely consumed during the first 48 h of fermentation independent of the genetic background, while maltose was consumed after 168 h in the HF strains. However, the LF strain stacked after 48 h (following glucose/fructose consumption), and we detected maltose consumption after 168 h (7 days). Overall, the LF strain consumed less than 50% of the maltose after 336 h, resulting in reduced fermentation capacity and ethanol production (Fig. 1D; Table S2C).

Similarly, we estimated the yeast assimilable nitrogen consumption (YAN) for the HF and LF strains at different time points during the fermentation process (24, 48, and

336 h, Table S2H). YAN was almost completely consumed after only 48 h; the CBS12357 strain had slower consumption kinetics compared with the QC18 and CL467.1 strains, mainly for ammonium (Student's *t* test, $P < 0.0001$) and some amino acids during the first 24 h (alanine, $P < 0.0001$; phenylalanine, $P = 0.0484$; and leucine, $P = 0.0095$; Student's *t* test, Table S2H), and for total YAN amino acids at 48 and 336 h (Student's *t* test, $P = 0.0076$ and $<0.0001$, respectively). These results demonstrate that beer fermentation differences across *S. eubayanus* strains are not due to differences in nitrogen consumption, but rather in maltose consumption.

**Differences in the glucose-maltose shift between *S. eubayanus* strains.** To evaluate the diauxic shift capacity of the HF and LF strains, particularly in the switch from consumption of glucose to other saccharides, we estimated their growth capacity under glucose, maltose, galactose, and sucrose after two 24-h precultures in 5% glucose (Fig. 2; Fig. S2; Table S3). In glucose (without diauxic shift), the growth kinetics profiles were similar between strains, yet the lag phase was shorter in the LF strain (Student's *t* test, $P = 3.9 \times 10^{-7}$, Fig. 2A). In contrast, we observed a significant difference between the QC18 and HF strains in their growth rates during the glucose-maltose shift (Student's *t* test, $P = 6 \times 10^{-9}$, Fig. 2B). This difference was only found for the diauxic shift between glucose and maltose; no differences were observed during the glucose-galactose or glucose-sucrose transitions (Fig. S2) The QC18 strain showed the largest glucose-maltose growth decrease, with a 94% decrease in growth rate and an 89% increase in the duration of the lag phase, compared to HF strains, which exhibited a 31% and 50% decrease in growth rate and a 17% and 22% increase in lag phase duration for the CBS12357 and CL467.1 strains, respectively. When a similar experiment was performed in maltose-maltose conditions, we found lower differences compared to the glucose-maltose shift, with a decrease of 31% in growth rate, and an increase of 41% for the lag phase in the LF strain, while the HF strains exhibited almost no differences in growth rates compared to the glucose-glucose condition (Fig. 2C). These results demonstrate that the differences in the glucose-maltose diauxic shift are responsible for the contrasting fermentation profiles between strains.

**Comparative transcriptomic analysis reveals transcription factors underlying fermentation differences between strains.** To explore global gene expression patterns that could explain fermentation differences between HF and LF strains, we performed RNA-seq analysis on samples collected 24 h after the beginning of the fermentation. This time point represents the inflection point when cells switch from glucose to maltose (8). We identified 418, 221, and 376 differentially expressed genes (DEGs) between CBS12357 versus QC18, CL467.1 versus QC18, and CL467.1 versus CBS12357, respectively (adjusted $P < 0.01$ and an absolute value of fold change >2, Fig. 3A; Table S4A). We identified a set of 93 DEGs in common between HF versus the LF strain, which could be related to differences in fermentation capacity. Using hierarchical clustering between HF strains versus the QC18 strain, we identified six clusters of expression profiles (Fig. 3B; Table S4B).

Interestingly, cluster I contained genes related to maltose metabolism and fatty acid beta oxidation, such as *MAL31*, *MAL32*, *IMA1*, *DCI1*, *FOX2*, and *PXA2* (Table S4B and S4C), which were upregulated in both HF strains. Cluster II contained upregulated genes in the QC18 strains related to transmembrane transport (*HXT5*, *QDR2*, and *ZRT1*), cell division (*CDC6* and *CLN2*), and methionine metabolic processes (*MET10*, *MET17*, *SAM2*, and *SAM3*). HF strains also differed in their expression patterns. Clusters III to VI showed different expression levels between both HF strains. For example, cluster III contained genes related to carbohydrate transport upregulated in CBS12357 and downregulated in CL467.1 and QC18. On the other hand, cluster IV contained genes related to response to stress and fungal-type cell wall organization downregulated in CBS12357 and upregulated in the other two strains (Fig. 2; Table S4B and S4C). These results suggest different regulatory mechanisms and molecular responses are needed to achieve high fermentation levels. We also performed a PCA of DEGs to analyze the expression patterns across the three strains (Fig. S3A), where the PC1 and PC2 components explain 62% and 35% of the observed variance, together accounting for 97% of the overall variation. PCA showed a separation of the three strains in both components;

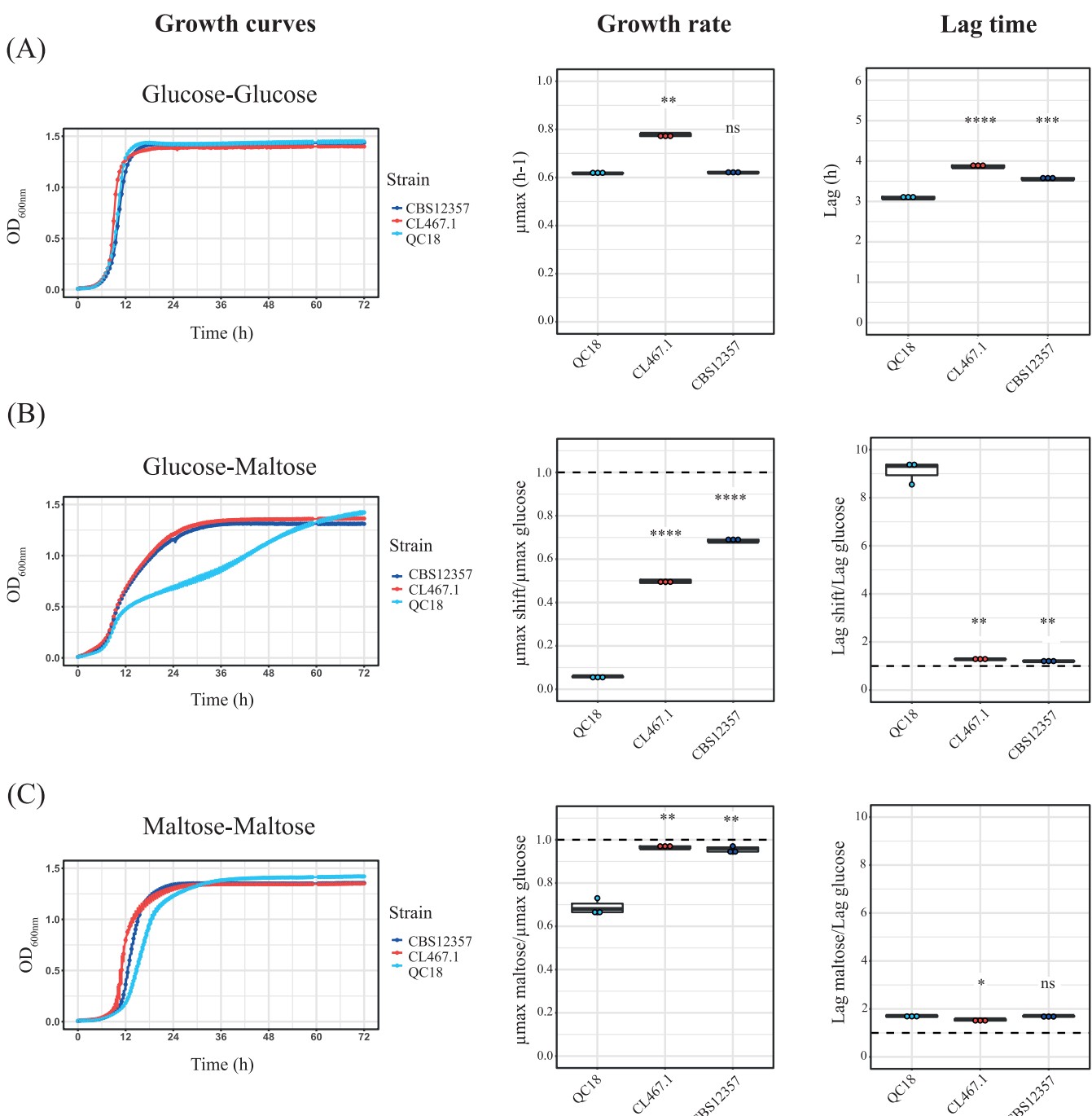

**FIG 2** Diauxic shift differences between LF and HF strains during the glucose-maltose shift. (A) Growth curves in glucose and the kinetic parameters: growth rate and lag time. (B) Growth curves in glucose-maltose shifts and the kinetic parameters relative to growth in glucose. (C) Growth curves in maltose and the kinetic parameters relative to the growth in glucose. Plotted values correspond to the mean value of three independent replicates for each strain. *, $P \leq 0.05$, **, $P \leq 0.01$, ***, $P \leq 0.001$, ****, $P \leq 0.0001$, different levels of significance between QC18 (LF) and the other strains (CBS12357 and CL467.1, HF) by *t* test.

the first component separated the CBS12357 strain from the other two strains, while the second component separated the CL467.1 strain from the other two strains. Although HF strains exhibited similar $CO_2$ loss profiles during the fermentation process, cluster analysis suggests different molecular responses in beer wort across all strains.

We then searched the YeTFaSco database to predict which transcription factors (TFs) potentially regulate DEGs (35). We found 76 TFs regulating DEGs between CBS12357 and QC18 (Table S4D), 43 TFs regulating DEGs between CL461.1 and QC18

(A)

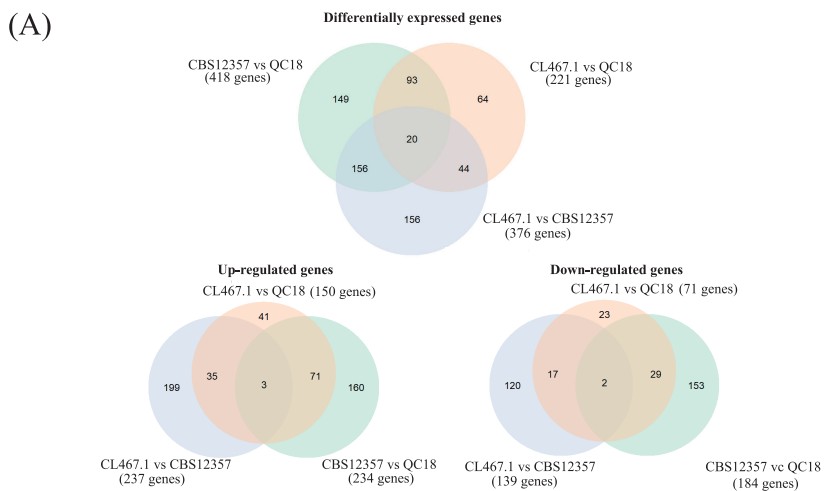

(B)

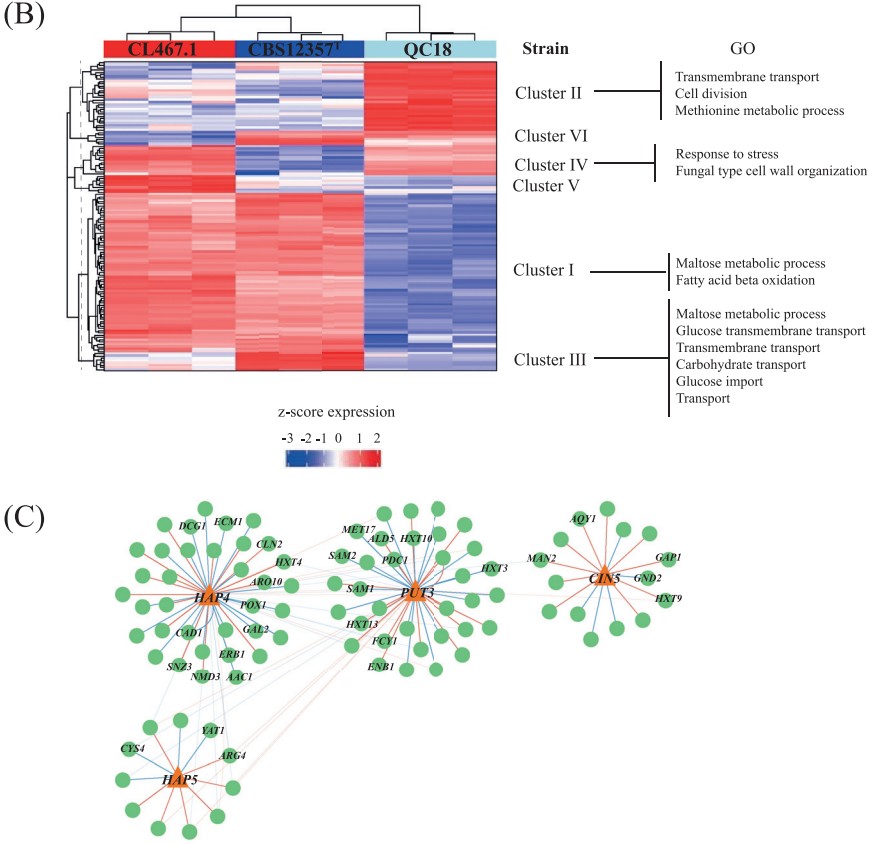

(C)

**FIG 3** Comparative transcriptome analysis between HF and LF strains. (A) Venn diagram of differentially expressed genes (DEGs), and up and downregulated genes in CBS12357 (HF) versus QC18 (LF), CL467.1 (HF) versus QC18, and CL467.1 versus CBS12357 strains. (B) Hierarchical clustering of DEGs in the three strains. The heatmap was generated using the z-score of expression levels in each comparison. Each row represents a given gene and each column represents a replica from a different strain. Clusters are annotated at the right, together with their Gene Ontology (GO) category. (C) Network analysis of DEGs regulated by Hap4p, Put3p, Cin5p, and Hap5 in strains with high fermentation capacity versus QC18 strain, depicting in bold the most relevant hubs. Red and blue lines represent positive and negative correlations, respectively.

(Table S4E), and 129 TFs regulating DEGs between CL467.1 and CBS12357 (Table S4F). To further identify transcriptional regulators underlying the fermentative differences between the three strains, we selected TFs based on four different criteria across strains: (i) significant binding differences prediction in DEGs, (ii) the presence of polymorphisms in the coding regions, (iii) differences in expression levels under fermentation conditions, and (iv) literature supporting their role in diauxic shift and stress responses during fermentation. In this way, we identified four TFs: Hap4p, Hap5, Put3p, and Cin5p. TFs belonging to the Hap complex (Hap2p, Hap3p, Hap4p, and Hap5p), a global regulator of respiratory gene expression and involved in diauxic shifts (36), were identified in the three comparisons, where Hap5p and Hap4p contain DNA-binding and DNA-activation domains, respectively (36). In addition, Cin5p is a member of the YAP family related to salt and osmotic tolerance (37), while Put3p regulates proline utilization genes (38); both TFs were identified in the comparisons of the two different sets of strains. An interaction network analysis for these four TFs (Fig. 3C) highlighted DEGs in common between HF strains versus QC18, pinpointing genes related to respiration (*AAC1*, *ATP1*, *ALD5*, and *YAT1*), nitrogen metabolism (*ARO10*, *DCG1*, *SNZ3*, *FCY1*, *MET17*, *SAM1*, *SAM2*, *GAP1*, *ARG4*, and *CYS4*), iron metabolism (*CAD1* and *ENB1*), cell cycle (*CLN2*), translation (*ECM1*, *ERB1*, and *NMD3*), hexose transport (*GAL2*, *HXT4*, *HXT10*, *HXT13*, *HXT3*, and *HXT9*), and fatty acid metabolism (*POX1*).

Between the CBS12357 and QC18 strains, the four chosen TFs harbor 5, 2, 0, and 2 nonsynonymous single nucleotide polymorphisms (SNPs) for *CIN5*, *HAP4*, *HAP5*, and *PUT3* (Table S4G), respectively. However, none of the amino acid substitutions were identified as deleterious to protein function (Table S4H). These results suggest that differences in expression levels can likely be explained by differences in polymorphisms within the regulatory regions of the target genes.

**Differences in chromatin accessibility and transcription factor binding between *S. eubayanus* strains.** We obtained transcription factor binding profiles in all three strains by performing ATAC-seq and measuring chromatin accessibility at promoters. Samples for ATAC-seq were collected after 20 and 72 h of wort fermentation to evaluate the chromatin accessibility during the diauxic shift. These time points represent the pre- and postglucose-maltose switch, since glucose is consumed in the first 24 h of the fermentation, while maltose consumption starts after 48 h. First, we analyzed differences in chromatin accessibility at 20 h and compared these to our gene expression results. We found 75 promoters that exhibited differential accessibility among strains (false discovery rate [FDR] <0.1) (Table S4I and S4J). We analyzed these differences using hierarchical clustering and identified four clusters (Fig. 4A). The largest cluster (cluster I) contained promoters showing higher accessibility in HF strains, and low accessibility in QC18. For example, the QC18 strain had lower accessibility and gene expression levels for the hexose transporter (*HXT9*), an activating protein of *CIN4* (*CIN2*), an iso-maltase (*IMA1*), and *GPB1*, a regulator of cAMP-PKA signaling that is involved in the glucose-mediated signaling pathway. In addition, in cluster II, which grouped promoters showing higher accessibility in CBS12357, we found MAL31 and MAL32, suggesting that the higher expression of these genes in CBS12357 could relate to their chromatin configuration. Furthermore, another copy of *MAL32* possessed higher accessibility in QC18, despite showing lower expression in this strain, likely suggesting a role for transcriptional repressors regulating *MAL32* expression in QC18.

HF strains exhibited differences between their chromatin accessibility patterns, represented in clusters II and III (Fig. 4A). In cluster II, we observed promoters showing higher accessibility in CBS12357, including genes related to maltose metabolism (*MAL31*, *MAL32*, and *IMA1*), transporters (*VBA5* and *HXT10*), and ion homeostasis (*FET4* and *ENB1*). In cluster III, we identified promoter regions with higher accessibility in CL467.1; however, these genes were unrelated to a particular metabolism function.

To increase our understanding of the regulatory differences occurring between these strains after the glucose to maltose shift, we analyzed chromatin accessibility differences after 72 h of fermentation. We found across all strains a total of 966 and 516 promoters with increased or decreased accessibility, respectively, when contrasting 72

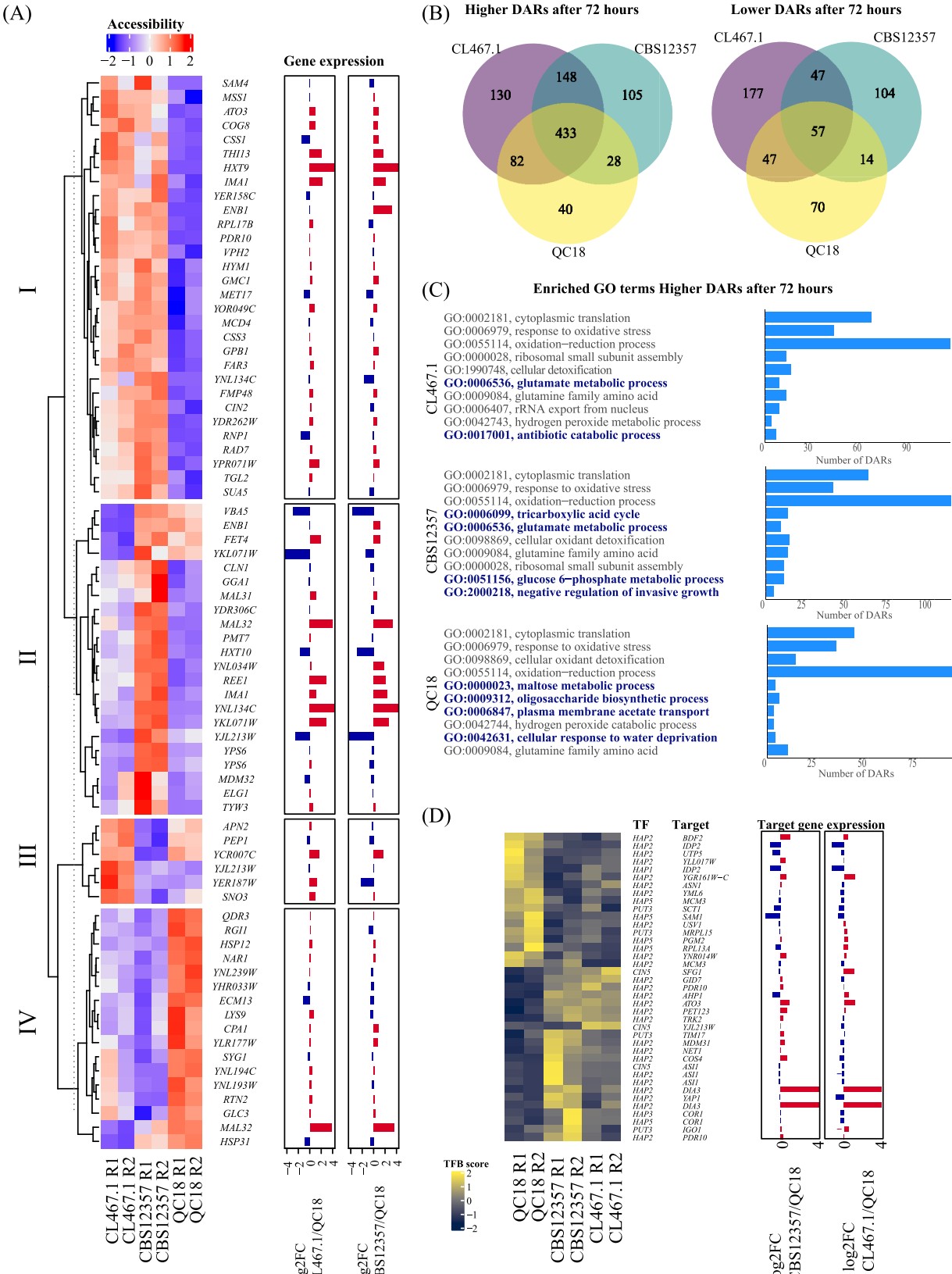

**FIG 4** Differences in chromatin accessibility and transcription factor binding between HF and LF strains. (A) The heatmap shows a hierarchical clustering analysis of chromatin accessibility at promoter regions of HF (CBS12357 and CL467.1) and LF (QC18) strains 20 h after fermentation. Accessibility from ATAC-seq fragments per kilobase per million values was transformed to z-scores and normalized by row. Gene expression at

to 20 h of fermentation (FDR <0.05). By comparing the sets of differentially accessible regions (DARs) between strains, we found that HF strains shared more DARs that increased accessibility after 72 h than with the LF strain (Fig. 4B). Gene ontology analyses of DARs with higher accessibility after 72 h highlighted similarities and differences between strains, where processes such as cytoplasmic translation and oxidative stress responses had increased accessibility in all strains (Fig. 4C). In contrast, processes that differed between strains included glutamate metabolism, which was more accessible in HF strains, and maltose metabolism, which was more accessible in the LF strain (Fig. 4C). These results likely suggest a delayed response in maltose consumption in the LF strain, compared to the HF strains.

Next, by profiling transcription factor binding footprints from ATAC-seq data, we explored the Hap complex, together with Cin5p, and Putp3 binding differences among strains. When examining genome-wide overall transcription factor binding scores (TFBS), we did not find significant differences between strains at 20 or 72 h of fermentation (Fig. S3B), suggesting that these TFs showed similar overall activity among strains. In addition, all differentially bound *Cin5p* binding sites ($n = 3$) showed higher TFBS in HF strains compared with QC18, with one of these sites associated with lower gene expression in the QC18 strain for *SFG1*, a putative transcription factor involved in the regulation of the mitotic cell cycle (39) (Fig. 4D). Interestingly, when dissecting TFBS variation at promoters, we found differences between strains mostly at Hap binding sites, with 25 differentially bound sites for Hap2p (Fig. 4D). Among others, we found lower expression levels and greater TFBS in QC18 for *IDP2*, a cytosolic NADP-specific isocitrate dehydrogenase with low levels in the presence of glucose (40), and *SAM1*, an *S*-adenosylmethionine synthetase, which promotes efficient fermentation (41). Overall, these results demonstrate how chromatin accessibility and TFBS differences promote significant differences between strains during the first stages of the beer fermentation process, which impact the fermentative and sugar consumption profile of the strains.

**HAP4, HAP5, and CIN5 impact the diauxic shift capacity in the LF strain.** In order to evaluate the effect of Hap4p, Hap5p, Cin5p, and Put3p on the diauxic shift and fermentation capacity in the LF and HF strains, we generated null mutants for these four TFs using CRISPR-Cas9 methodology (Fig. 5A; Fig. S4A, S4B, and S4C; Table S5A). In HF strains, the *hap4Δ*, *hap5Δ*, and *cin5Δ* knockouts showed a lengthening of the lag phase and an increase in their growth rate after the glucose-maltose shift. In contrast, these knockouts in the QC18 strain exerted a different effect by decreasing the duration of the lag phase and increasing the growth rate, with the greatest effect observed in the *cin5Δ* knockout (~14-fold increase in growth rate versus the wild-type strain; Student's *t* test, $P = 3.045 \times 10^{-8}$). These results suggest that Cin5p could strongly activate a repressor of the diauxic shift in the LF strain but not necessarily in the HF strains.

Since *cin5Δ* and *hap5Δ* mutants exerted the strongest phenotypes, we generated double mutants for *cin5Δ/hap5* combinations. Interestingly, in the QC18 strain, the *cin5Δ/hap5* double mutant did not exhibit any differences in terms of lag phase duration, and it possessed a growth rate value between that of the *cin5Δ* and the *hap5* single mutant strains. The equivalent double mutants in the HF strains exhibited a decrease in the lag duration (Student's *t* test, $P = 0.02117$ and $7.1 \times 10^{-6}$ for CBS12357 and CL467.1 strains, respectively) and a mild increase in their growth rate (Student's *t* test, $P = 1.6 \times 10^{-4}$ and $6.8 \times 10^{-6}$ for CBS12357 and CL467.1 strains, respectively), relative to the wild-type strains. For the *put3Δ* null mutants, statistically significant differ-

**FIG 4** Legend (Continued)
20 h is shown as $\log_{2\text{-fold}}$ changes of CL467.1 and CBS12357 relative to QC18. (B) The Venn diagram shows the number of differentially accessible regions (DARs) and their intersection showing higher or lower accessibility in the HF and LF strains in the contrast between 72 and 20 h of fermentation. (C) Gene ontology (GO) enrichment analyses for DARs highlight higher accessibility after 72 h of fermentation. GO terms correspond to Biological Processes. (D) The heatmap shows transcription factor (TF) binding scores obtained from ATAC-seq TF binding footprints transformed to z-scores and normalized by row. Gene expression is shown as in panel A.

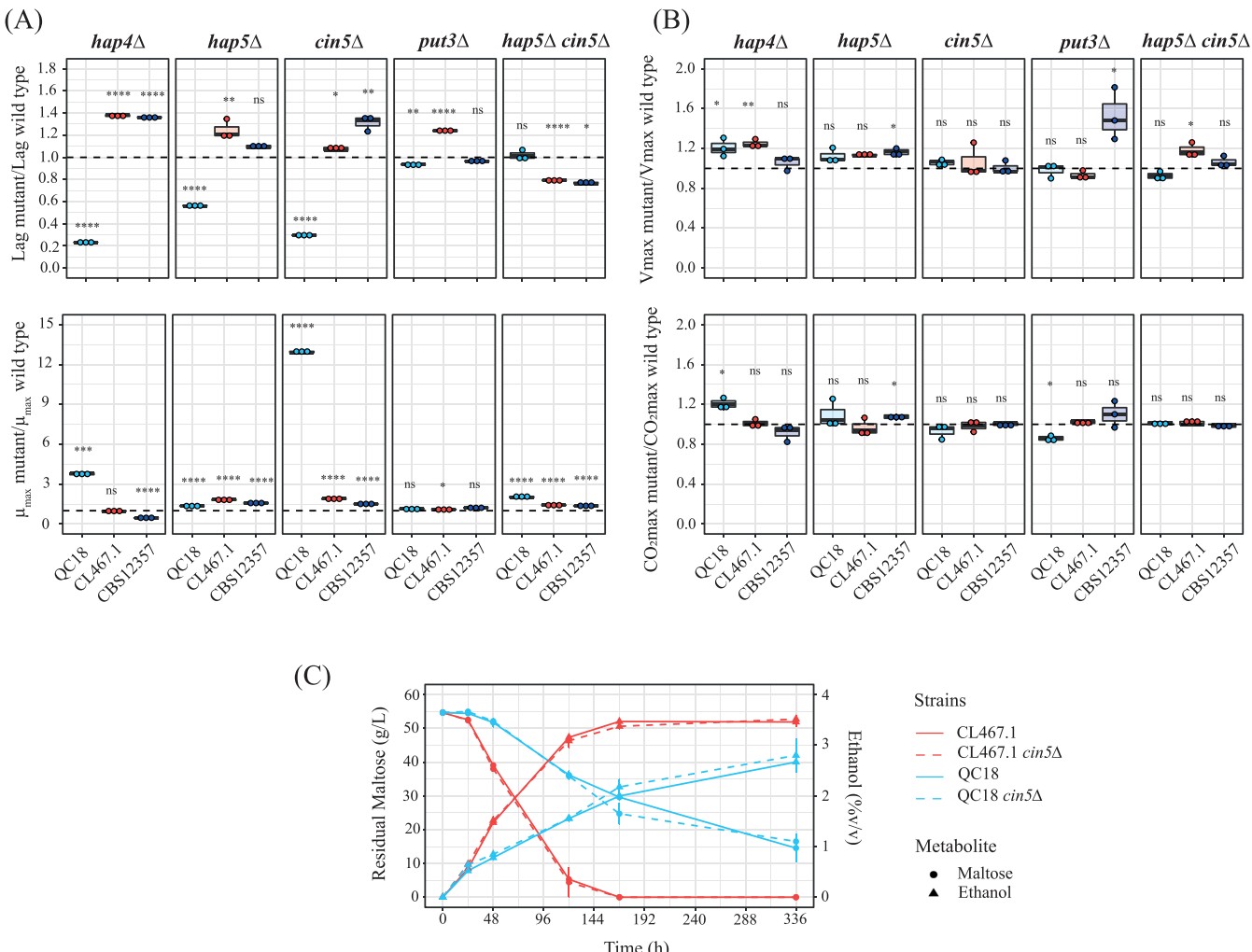

**FIG 5** Effect of Hap4p, Hap5, Cin5p, and Put3p on diauxic shift and fermentation capacity. (A) Lag time and growth rate of null mutant strains relative to wild-type strains after glucose-maltose shift. (B) Fermentation rate and maximum $CO_2$ loss of null mutant strains relative to wild-type strains. (C) Maltose consumption and ethanol production kinetics of CL467.1 and QC18 *cin5* null mutant strains. Plotted values correspond to the means of three independent replicates of each strain. * $P \leq 0.05$, **, $P \leq 0.01$, ***, $P \leq 0.001$, ****, $P \leq 0.0001$, different levels of significance between mutant and wild-type strains (*t* test).

ences were observed solely for the QC18 and CL467.1 strains, with decreases and increases in the duration of the lag phase, respectively (Fig. 5A, top panels), while no differences were observed for growth rate.

We evaluated the effect of mutating these four TFs on the diauxic shift for other sugar sources, such as glucose-galactose and glucose-sucrose (Fig. S4B and S4C). In these cases, no differences in the lag phase duration for QC18 mutants after the glucose-galactose shift were observed, except for the double mutant that suffered an increase (Student's *t* test, $P = 0.0003$, Fig. S4B, top panels). Furthermore, we detected a rise in the growth rate in almost all mutants (Fig. S4B, bottom panels). The glucose-sucrose shift exerted a different pattern. Here, we noted an increase in the lag phase duration and a decrease in the growth rate for the *cin5Δ* and *hap5* QC18 mutants. The four TFs also impacted the fermentation capacity in the three strains under study (Fig. 5B; Table S5B). Knockout of *HAP4* affected the $CO_2$ loss rate in QC18 and CL467.1 strains, and the maximum $CO_2$ loss in the QC18 strain, increasing the fermentation capacity in comparison with their respective wild-type strains. A similar effect was observed in the *hap5Δ* knockout of the CBS12357 strain. These results suggest that the TFs Hap4p, Hap5p, and Cin5p participate in the diauxic shift between different carbon sources in *S. eubayanus*, but

their effect is dependent on the genetic background and the disaccharide carbon source.

To increase our understating of the allelic differences between the three strains during the diauxic shift, we determined the effect of *HAP5* and *CIN5* allelic variants, the two TFs with the greatest effects, by performing a reciprocal hemizygosity analysis between the QC18 and the HF strains (Fig. S4D; Table S5C). In this assay, we only observed differences between the *CIN5* hemizygous strains, where the *CIN5*-QC18 allele had a lower growth rate compared to the CL467.1 allele (Fig. S4D), and a greater lag phase compared to the CBS12357 allele (Fig. S4D), demonstrating a significant effect depending on the *CIN5* allelic version. Finally, we evaluated the kinetics of maltose consumption in *CIN5* null mutants of strains CL467.1 and QC18 under fermentative conditions. Although we did not observe statistically significant differences during fermentation, the null mutant in QC18 showed a mild tendency to higher maltose consumption after 120 h (Student's $t$ test, $P = 0.05$, Fig. 5C; Table S5D). Altogether, our results demonstrate that Cin5p significantly affects the glucose-maltose shift in *S. eubayanus* and is dependent on the genetic background.

## DISCUSSION

The study of natural variation can identify novel alleles with immense value for the development of novel genetic stocks with applications in several fields. New alleles identified in yeast can improve the fermentation performance of lager hybrids and the generation of unique fermentative profiles in brewing. Until now, *S. eubayanus* has only been isolated from wild environments and has never been associated with anthropogenic niches. Still, *S. eubayanus* can grow and ferment malt extract and has a broad phenotypic diversity in terms of fermentative capacities and aroma compound production (4, 6–8, 42). Considering that Patagonian isolates have a greater global genetic diversity, known so far, than Holarctic and Chinese isolates (4, 5), these strains from the Southern Hemisphere are a rich genetic reservoir for the identification of allelic variants and genetic stocks for the generation of novel hybrids with brewing potential.

Most of the *S. eubayanus* isolates analyzed from Patagonia showed similar fermentation capacities when comparing the maximum $CO_2$ loss (yield). However, we found a lower fermentative capacity in all these strains compared to the *S. pastorianus* W34/70 commercial strain. These differences arise because *S. eubayanus* does not consume maltotriose, the second most abundant sugar present in beer wort (43). However, we identified a greater variability in the maximum $CO_2$ loss rate, mostly due to differences in maltose consumption. Indeed, fermentation rates are of major importance during alcoholic fermentation, mostly because strains able to rapidly ferment all the available sugars can take over the culture and inhibit the growth of other microorganisms (24, 44). Glucose promotes catabolite repression, such that maltose uptake usually begins only after half of the initial glucose concentration has been consumed (45). The sensitivity to glucose-induced inhibition is strain specific and the expression of genes responsible for maltose metabolism may be either induced or constitutive (45). Indeed, we found differences between strains in their ability to switch from glucose to maltose, and one of the strains was extremely slow in this transition (LF, QC18). Studies of wine fermentation have demonstrated that the slow fermentation of fructose is strain and time dependent (46). Similarly, the maltose consumption rate of yeast determines the fate of the brewing fermentation process, which differs across strains (47). Thus, consumption rate and residual maltose are of crucial relevance to the brewing industry.

The QC18 strain has the lowest maximum $CO_2$ loss and $CO_2$ loss rate values. This strain belongs to the PA cluster and was obtained in the Nahual Huapi National Park (Argentina), but unlike the other strains, QC18 was isolated from the bark of an exotic tree in Patagonia: *Quercus robur* (European oak) (6). Oaks are dominant woody species throughout the Northern Hemisphere (48), unlike trees belonging to *Nothofagus* genus, which usually form the core of the South Hemisphere primary native forests of Argentina, Chile, Australia, and New Zealand (49). The tree host could determine the

phenotypic differences between strains, likely due to differences in the complexity of the available sugars in the bark. In this sense, *Quercus* barks are mostly composed of saccharides such as glucose and xylose (50), while *Nothofagus pumilio* trees have higher concentrations of more complex sugar sources, such as starch (51). The QC18 strain has a poor capacity to switch from glucose to maltose, with a decrease in growth rate and an increase in lag time, compared to growth in glucose. There are undoubtedly several factors that may affect the adaptation of the QC18 strain to maltose. During the diauxic shift, there is reduced respiratory activity that lengthens the lag phase, while an overactivation of respiration results in shorter lag phases (25, 52). Upon the sudden loss of glucose, cells enter an energy-deficient state because they are not able to metabolize maltose (52). This energy-deficient state would be exacerbated by decreased respiratory activity in response to glucose. Finally, this state likely prevents induction of *MAL* and subsequent escape from the lag phase.

Many genes are influenced by carbon catabolite derepression during industrial brewery fermentations (53). Analysis of gene expression patterns across strains allowed us to identify strain-dependent mechanisms that explain the differences and similarities in fermentation profiles. Analysis of DEGs and chromatin accessibility showed different patterns among HF strains, demonstrating different molecular mechanisms toward convergent phenotypes, such as fermentation capacity. In particular, the CBS12357 strain contains upregulated genes related to carbohydrate transport, while CL467.1 upregulates genes related to the response to stress and fungal-type cell wall organization. Interestingly, we observed higher accessibility and higher gene expression levels in *MAL31* and *MAL32* in the CBS12357 strain compared to QC18, suggesting a reduced maltose activity in the latter strain. Still, further studies are needed to determine whether *MAL* genes are functional in QC18 strain. This reduced adaptation to the diauxic shift was only observed during the shift from glucose to maltose but not to sucrose or galactose. In agreement with these results, previous studies have demonstrated large differences between strains in gene expression changes and growth rates under different carbon sources, where the diauxic shift depends on both the yeast strain and the carbon source (52, 54). In addition, the CBS12357 strain exhibited greater accessibility in the promoter region of the *MAL* genes after 20 h, whereas the QC18 strain showed lower transcriptional activity of maltose metabolism genes and higher expression levels of *HXT5*, which encodes a functional hexose transporter with moderate affinity for glucose (55). This gene is regulated by growth rates rather than by extracellular glucose, particularly under conditions that cause slow cell growth, e.g., upon carbon and nitrogen starvation. *HXT5* transcription depends on two HAP complex binding elements and one postdiauxic shift element in its promoter region (27). The HAP complex (HapII/3/4/5) plays a central role in converting cells from fermentative to respiratory growth following the diauxic shift by inducing genes required for mitochondrial function upon glucose depletion (25, 52, 56). In this sense, our gene expression analysis identified four TFs that are significantly overrepresented across DEGs and that have been previously related to the fermentation process, including TFs belonging to the HAP complex. Previous reports demonstrated in the BY4742 haploid *S. cerevisiae* strain that null mutants of the HAP complex generate a lengthening in the duration of the lag phase (25). However, whether this phenotype is conserved across other isolates is unknown. Indeed, our results suggest that this pattern might not be conserved across isolates or even species in the *Saccharomyces* genus and alternative regulatory mechanisms might operate under different genetic backgrounds. While the HF knockout strains showed similar phenotypes to those described in *S. cerevisiae*, i.e., lag phase lengthening in HAP complex mutants (25), the QC18 strain showed a contrasting phenotype by significantly improving its growth rate in the glucose-maltose shift in *hap4Δ* and *hap5Δ* null mutants. Part of this alternative mechanism could be upstream of the HAP complex regulation. In this sense, our analysis also highlighted the role of Cin5p as responsible for differences across strains, where *cin5Δ* null mutants showed a shorter lag phase in HF strains. Cin5p is a basic leucine zipper TF of the yAP-1 family, which participates in several stress conditions, including oxidative and osmotic stress (57). These

results demonstrate the power of coupling RNA-seq with ATAC-seq to identify genetic variants responsible for phenotypic differences between different backgrounds. Alternative approaches, such as QTL mapping, can allow identifying large and small effect variants (58). However, this approach is more laborious and requires generating a large set of segregants, together with genotyping and phenotyping efforts. Instead, the analysis of the parental strains using two-cutting edge techniques, such as RNA-Seq and ATAC-Seq, rapidly identifies the genetic determinants and regulatory variants underlying the phenotypic variation between strains. Transcriptional regulation during the diauxic shift in yeast might be more complex than observed in other strains, and further studies are needed to elucidate the role of these genes and allelic variants during this transition.

In conclusion, the identification of natural allelic variants for fermentation rate is instrumental to develop novel strains for brewing. In this study, we identified three TFs, *HAP4*, *HAP5*, and *CIN5*, that are responsible for differences in the glucose-maltose diauxic shift between strains, with the most profound effect being present in *CIN5* allelic variants. In this context, HF and LF strains exhibited unique gene expression patterns during fermentation, with HF strains showing greater expression levels of maltose genes, while the LF strains of glucose-related transporters. These differences could originate in the tree host from which this strain was isolated, which might determine the sugar source preference. Novel alleles providing high fermentation rates will be of great value for the generation of novel lager hybrids in the brewing industry.

## MATERIALS AND METHODS

**Strains and culture media.** The *S. eubayanus* strains used in this work are listed in Table S1A and were collected from different Chilean and Argentinean localities (4–6). We also used the *S. pastorianus Saflager* W34/70 strain (Fermentis, France) as a lager fermentation control. All the strains were maintained on YPD solid media (1% yeast extract, 2% peptone, 2% glucose, 2% agar). For long-term storage, the strains were maintained at $-80°C$ in 20% glycerol.

**Fermentations in beer wort.** Fermentations were carried out in three biological replicates as previously described (33). Briefly, 12°Plato (°P) beer wort was oxygenated (15 mg $L^{-1}$) and supplemented with 0.3 ppm $Zn^{2+}$ (as $ZnCl_2$). The precultures were grown in 5 mL of 6°P wort for 24 h at 20°C in constant agitation at 150 rpm. Then, the inoculum was transferred to 50 mL 12°P wort and incubated for 24 h at 20°C in constant agitation at 150 rpm. The cells were collected by centrifugation at 5,000 $\times$ *g* for 5 min. The final cell concentration for each fermentation was estimated according to the formula described by White and Zainasheff (59). Cells were inoculated into 100 mL 12°P wort, using 250-mL bottles and airlocks with 30% glycerol. The fermentations were incubated at 12°C, with no agitation for 10 to 15 days, and monitored by weighing the bottles daily to determine weight loss over time. The maximum $CO_2$ loss rate ($V_{max}$) was estimated using the R software version 4.1.1. The $CO_2$ loss curves were smoothened and the first derivative was plotted using the smooth.spline function. The maximum $CO_2$ loss rate coincides with the maximum point of the first derivative.

**Metabolite quantification by high-performance liquid chromatography.** Sugar (glucose, fructose, maltose, and maltotriose) and nitrogen (ammonium and amino acids) consumption, together with glycerol and ethanol production, was determined by high-performance liquid chromatography at different time points during the fermentation process as previously described (8, 42, 60). These analyses were carried out in triplicate at 24 h, 48 h, and on the final day of fermentation, and the consumption of each nitrogen source was estimated as the difference between the initial value and that of each time point of the fermentation.

**RNA-seq analysis.** Gene expression analysis was performed on strains CL467.1, CBS12357, and QC18, which exhibited significant differences for different fermentative phenotypes. These strains were fermented in triplicate as previously described for 24 h. Then, cells were collected by centrifugation at 5,000 $\times$ *g* for 5 min and treated with 2 units of Zymolyase (Seikagaku Corporation, Japan) for 30 min at 37°C. RNA was extracted using the E.Z.N.A Total RNA kit I (OMEGA) according to the manufacturer's instructions and treated with DNase I (Promega) to remove genomic DNA traces. Total RNA was recovered using the GeneJET RNA Cleanup and Concentration Micro kit (Thermo Fisher Scientific). RNA integrity was confirmed using a Fragment Analyzer (Agilent). RNA library preparation and Illumina sequencing were performed in the BGI facilities (Hong Kong, China) as previously described (33).

The quality of the raw reads was evaluated using the fastqc (https://www.bioinformatics.babraham.ac.uk/projects/fastqc/) tool. Reads were processed using fastp (-3 l 40) (Mao and Chen [36]) and mapped against the *S. eubayanus* CBS12357 reference genome (28) using STAR (61). DEG analysis was performed using the DESeq2 package (62) in R v4.1.2, comparing the three strains at the same time. Genes with an adjusted $P < 0.01$ and an absolute value of fold change $>2$ were considered DEGs for each comparison (CL467.1 versus QC18, CBS12357 versus QC18, and CL467.1 versus CBS12357).

**ATAC-seq data analysis.** The assay for transposase accessible chromatin analysis (ATAC-seq) was performed on strains CL467.1, CBS12357, and QC18. These strains were fermented in duplicate and after 20 h and 72 h, 2.5 million cells were collected by centrifugation at 1,800 $\times$ *g* for 4 min at room

temperature and washed twice using SB buffer (1 M sorbitol, 10 mM MgCl$_2$, 40 mM HEPES pH 7.5). Then, cells were treated with 50 mg/mL zymolyase 20T (Seikagaku Corporation, Japan) in SB buffer for 30 min at 30°C. After incubation, cells were washed twice with SB buffer, resuspended in 50 $\mu$L transposition mix, containing 25 $\mu$L Nextera Tagment DNA buffer (Illumina, USA), 22.5 $\mu$L H$_2$O and 2.5 $\mu$L Nextera Tagment DNA enzyme I (Illumina, USA). After incubation for 30 min at 37°C, DNA was purified using the DNA Clean and Concentrator-5 kit (Zymo Research), according to the manufacturer's instructions. Tagmented DNA was amplified by PCR using 1$\times$ NEBNext Hi-Fidelity PCR Master Mix (New England Biolabs [NEB]), using Nextera Index i5 and i7 series PCR primers, and 5 $\mu$L tagmented DNA. Then, 50 $\mu$L of the amplified ATAC-seq library was subjected to double-sided size selection using magnetic beads (AMPure XP, Beckman Coulter). DNA bound to the beads was washed twice with 80% ethanol and then eluted in 20 $\mu$L H$_2$O. Library quality was assessed using a Fragment Analyzer (Agilent, USA) and quantified in Qubit (Thermofisher, USA). Sequencing was conducted on a Nextseq 500 (Illumina, USA) in the Genomics unit at Universidad de Santiago de Chile. ATAC-seq reads were analyzed as previously described (60).

To match genes with their nearby ATAC-seq signal, we selected a regulatory region of 400 bp upstream of the start codon site for each gene. The ATAC-seq signals of 5,433 regulatory regions were quantified by counting mapped reads using featureCounts. Differential responses in ATAC-seq were estimated using DESeq2 (design= $\sim$ condition).

Transcription factor binding scores (TFBSs) were calculated using TOBIAS (63) and the ScoreBigWig tool (–fp-min 5 –fp-max 30) for 141 yeast TFs from the JASPAR database (64). Binding scores were further processed in R. To calculate statistical differences in TFBSs, we employed a linear model using the limma R package (65). We considered binding differences with a FDR $<$0.1 as statistically significant.

**Gene ontology analysis.** Gene ontology analysis was performed using the tools provided by the DAVID Bioinformatics Resource (66, 67) using DEGs with $P < 0.01$ and absolute fold change values $>2$. We selected categories with a significant overrepresentation utilizing a FDR $<$10%.

**Transcription factor analysis.** Correlations between TFs and DEGs were predicted utilizing YeTFaSCo: Yeast Transcription Factor Specificity Compendium (35). Fold change values were used to test associations with potential regulators using a Spearman correlation.

The open reading frame sequences of the selected TFs for strains CL467.1 and QC18 were obtained directly from the BAM files (.bam) mapped against the reference genome. BAM files were converted to VCF (Variant Calling Format) files using *freebayes*, which contained the genotype information of each gene and strain (68). Then, FASTA files were generated from VCF files using SAMtools (69). The nucleotide sequences were translated into amino acid sequences in Geneious v 8.1.8, utilizing the standard genetic code. Amino acids and nucleotide sequences were then aligned against the reference strain (CBS12357) using a Multiple Comparison by Log-Expectation (MUSCLE) algorithm with default parameters in Geneious v 8.1.8. The prediction of the change of each nonsynonymous SNP over the protein sequence was analyzed using the online tool PROVEAN v1.1.3 (70).

**Coexpression networks analyses.** Coexpression networks were generated using a subset of the whole DEG set regulated by the TFs Hap4p, Hap5p, Cin5p, and Put3p, as previously described (71). For this, we used read counts to calculate gene expression correlation. First, read counts were normalized to the number of reads that were effectively mapped, using the median normalization method available in the EBSeq R package (72). Subsequently, we added pseudocounts to avoid values equal to zero, generating a logarithmic matrix of the data. Finally, we used the Spearman correlation between each pair of selected groups of DEGs using the "psych" R package, retaining correlations with absolute values $>0.9$ and adjusted $P$ values $<0.05$. The network statistics, "degree" and "betweenness centrality," were calculated using "igraph" (https://igraph.org/). Cytoscape v 3.9.1 (73) was used to visualize networks and corresponding statistics.

**Generation of null mutants and reciprocal hemizygote strains.** Null mutants for the *HAP5*, *HAP4*, *PUT3*, and *CIN5* genes were generated using the CRISPR-Cas9 method (74) as previously described (9). Briefly, the gRNAs were designed using the Benchling online tool (https://www.benchling.com/) and cloned in the plasmid pAEF5 (a gift from Gilles Fischer, Addgene plasmid no. 136305) (75), using standard "Golden Gate Assembly" (76). CL467.1, CBS12357 and QC18 strains were cotransformed with the plasmid carrying the gRNA and the Cas9 gene and with a synthetic double-stranded DNA fragment (donor DNA) composed of a 100-bp sequence containing flanking sequences of the target gene, corresponding to 50-bp upstream of start codon and 50-bp downstream of the stop codon. Correct gene deletion was confirmed by standard colony PCR. All the primers, gRNAs, and donor DNA are listed in Table S6. Double null mutants for the TFs *HAP5* and *CIN5* were generated using the CRIPSR-Cas9 method as described above. The second null deletion was performed in Δ*hap5* strains. We were unable to cure Δ*hap4* strains and to generate second null deletions in this background.

Reciprocal hemizygote strains were generated for *HAP5* and *CIN5* genes. The diploid wild-type strains CL467.1, CBS12357, QC18, and their respective null mutant strains were sporulated on 2% potassium acetate agar plates (2% agar) for at least 7 days at 12°C. Meiotic segregants were obtained by dissecting tetrad ascospores treated with 10 $\mu$L Zymolyase 100T (50 mg/mL) in a SporePlay micromanipulator (Singer Instruments, UK) and crossed on YPD agar plates. Crossbreeding corresponds to crossing one wild-type strain against another containing the mutated target gene. The plates were incubated at 25°C for 3 days, and then colonies were isolated and checked for the correct genotype by colony PCR. All the strains generated are listed in Table S1B.

**Phenotypic characterization of strains.** Wild-type strains, null mutants, and reciprocal hemizygote strains were phenotypically characterized under fermentation and/or microculture conditions as previously described (9, 33). Diauxic shift experiments were performed as previously published (33). Briefly,

precultures were grown in YP (1% yeast extract, 2% peptone) containing 5% glucose medium at 25°C for 24 h. Cultures were diluted to an initial optical density at 600 nm ($OD_{600nm}$) of 0.1 in fresh YP 5% glucose medium for extra growth overnight. The next day, cultures were used to inoculate a 96-well plate with a final volume of 200 $\mu$L YP with the carbon source (5% glucose, 5% maltose, 5% galactose, or 5% sucrose) at an initial $OD_{600nm}$ of 0.1. The growth curves were monitored by measuring $OD_{600nm}$ every 30 min in a TECAN Sunrise instrument. Lag phase and $\mu_{max}$ were estimated as previously described (25) using the R software version 4.1.2.

**Data and statistical analyses.** Statistical analysis and data visualization were performed using the R software version 4.1.2. The fermentation and growth kinetic parameters were compared using ANOVA and the mean values of the three replicates were statistically analyzed with a Student's *t* test and corrected for multiple comparisons using the Benjamini-Hochberg method. $P < 0.05$ was considered statistically significant. PCA was performed using the FactoMineR package version 2.4 to compute principal component methods and the factoextra package version 1.07 for extracting, visualizing, and interpreting the results. Heatmaps were generated using the ComplexHeatmap package version 2.6.2.

**Data accessibility.** All sequences from RNA-seq and ATAC-seq have been deposited in the National Center for Biotechnology Information as a Sequence Read Archive under the BioProject accession number PRJNA857309.

## SUPPLEMENTAL MATERIAL

Supplemental material is available online only.
**FIG S1**, PDF file, 0.6 MB.
**FIG S2**, PDF file, 0.5 MB.
**FIG S3**, PDF file, 0.3 MB.
**FIG S4**, PDF file, 0.6 MB.
**TABLE S1**, XLSX file, 0.01 MB.
**TABLE S2**, XLSX file, 0.1 MB.
**TABLE S3**, XLSX file, 0.01 MB.
**TABLE S4**, XLSX file, 1.6 MB.
**TABLE S5**, XLSX file, 0.02 MB.
**TABLE S6**, XLSX file, 0.01 MB.

## ACKNOWLEDGMENTS

We thank Kamila Urbina, Antonio Molina, and Jaime Ortega for their technical help and Michael Handford (Universidad de Chile) for language support. F.A.C. acknowledges the Comisión Nacional de Investigación Científica y Tecnológica CONICYT FONDECYT (1220026) and ANID–Programa Iniciativa Científica Milenio–ICN17_022 and NCN2021_050. J.M. is supported by FONDECYT POSTDOCTORADO grant no. 3200545 and P.V. by FONDECYT POSTDOCTORADO grant no. 3200575. R.F.N. is supported by FIC "Transferencia Levaduras Nativas para Cerveza Artesanal" and FONDECYT grant no. 1180917. J.B.-P. is supported by Proyecto Dicyt Postdoc/ayudante 022243CR_Postdoc, Vicerrectoría de Investigación, Desarrollo e Innovación. D.L. is supported by CONICET project PIP 11220200102948CO, ANPCyT project PICT 2020-00226, and Universidad Nacional de Comahue. This research was partially supported by the supercomputing infrastructure of the National Laboratory for High Performance Computing Chile (NLHPC, ECM-02). We also acknowledge Fundación Ciencia & Vida for providing infrastructure, laboratory space, and equipment for experiments. The funders had no role in study design, data collection and interpretation, or the decision to submit the work for publication.

Conceptualization, J.M., J.I.E., D.L., and F.A.C.; Methodology, J.M., J.I.E., P.Q., D.L., and F.A.C.; Software, J.M., C.A.V., N.B. and P.V.; Validation, J.M., R.F.N., D.L., and F.A.C.; Formal Analysis, J.M., P.Q., N.B., and C.A.V.; Investigation, J.M., J.I.E., P.Q., P.V., J.B.-P., and C.A.V.; Resources, J.M., P.V., R.F.N., D.L. and F.A.C.; Visualization, J.M., P.Q., and C.A.V.; Data Curation, J.M., P.Q., C.A.V., and N.B.; Writing – Original Draft Preparation, J.M., J.I.E., C.A.V., and F.A.C. All authors have read and agreed to the published version of the manuscript.

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
