## [Reviewer comments · mSystems]

Natural Variation in Diauxic Shift between Patagonian *Saccharomyces eubayanus* Strains

Jennifer Molinet, Juan Eizaguirre, Pablo Quintrel, Nicolas Bellora, Carlos Villarroel, Pablo Villarreal, Jose Benavides-Parra, Roberto Nespolo, Diego Libkind, and Francisco Cubillos

Corresponding Author(s): Francisco Cubillos, Universidad de Santiago de Chile

Review Timeline:

Submission Date:	July 9, 2022
Editorial Decision:	September 12, 2022
Revision Received:	October 5, 2022
Editorial Decision:	November 10, 2022
Revision Received:	November 11, 2022
Accepted:	November 16, 2022

Editor: Benjamin Wolfe

Reviewer(s): Disclosure of reviewer identity is with reference to reviewer comments included in decision letter(s). The following individuals involved in review of your submission have agreed to reveal their identity: Jean-Luc Legras (Reviewer #2)

Transaction Report:

DOI: <https://doi.org/10.1128/msystems.00640-22>

September 12, 2022

Dr. Francisco A Cubillos
Universidad de Santiago de Chile
Santiago
Chile

Re: mSystems00640-22 (Natural Variation in Diauxic Shift between Patagonian *Saccharomyces eubayanus* Strains)

Dear Dr. Francisco A Cubillos:

Thank you for submitting your manuscript to mSystems. We have completed our review of your manuscript. Both reviewers found many strengths in your manuscript. But they also pointed out some minor issues that need to be addressed in a revised manuscript. Below you will find instructions from the mSystems editorial office about how to submit a revised manuscript as well as comments from the reviewers.

Preparing Revision Guidelines

Sincerely,

Benjamin Wolfe

Senior Editor, mSystems

Journals Department
Reviewer comments:

Reviewer #1 (Comments for the Author):

Molinet et al investigate strain differences in fermentation profiles of wild isolates of *S. eubayanus*. They characterize differential expression and chromatin accessibility in two high ferment strains and one low ferment strain. They identify groups of genes that are differentially expressed underlying the diauxic shift from glucose to maltose, a process that is important in fermentation. They identify four important transcription factors and create null mutants in relevant backgrounds, showing that CIN5 is a repressor of diauxic shift in the low ferment strain. However the CIN5 effect seems to be dependent on the carbon source and the genetic background, so while these results are really interesting from an evolutionary perspective, how this might be translational is unclear. Nevertheless, I thought overall this was a clear and comprehensive study.

A further elaboration of phenotypes in *S. cerevisiae* (and/or *S. pastorianus*) and how they compare with *S. eubayanus* both in terms of fermentation phenotypes, diauxic shift, and transcription factor mutants would provide helpful context. I'm assuming that diauxic shift from glucose to maltose is well described in *S. cerevisiae* (but perhaps not incorporating strain diversity?). What does this new work add to the field, and what can you say about how the key players have diverged or been conserved functionally?

The main conclusions of this study focus more on what's wrong with the low fermenting strain (QC18) than on what is better about the high fermenting strains, and it's unclear to me if the high fermenting strains are actually high fermenting, or just relative to the poor fermenting strain. In this respect, I think that the variation between the 2 high fermenting strains analyzed is an interesting path for further exploration for alleles that may be relevant for brewing strain improvement, since the authors showed that a similar fermentation phenotype may be regulated by different molecular mechanisms. This could use further discussion.

I thought there were several points of disconnect about identifying important alleles for fermentation improvement, when really the only alleles discussed was the QC18 CIN5 allele and it was in a very specific context of a poor fermenting strain under non-fermenting conditions.

What is the architecture of MAL genes in *S. eubayanus* - how many MAL gene clusters are there? Is there copy number variation? Could certain alleles be non-functional in QC18?

Lines 251-263: Perhaps this section could be reworded and clarified to make it more clear why these transcription factors were selected, especially because this dictates the remaining experiments.

Lines 264-265: It would be useful to highlight here that all but 1 nonsynonymous difference are between the QC18 strain and the reference. Allelic differences do seem to be important (as described later in the hemizyosity tests). Is the QC18 CIN5 allele found in any other *S. eubayanus* strains?

Line 426: glucose is not a polysaccharide

Reviewer #2 (Comments for the Author):

In this manuscript, Molinet et al. study the mechanisms explaining the different abilities of South American *S. eubayanus* strains to ferment and switch from glucose to maltose.

Starting from the comparison of 19 strains, the authors retained two strains fermenting quickly maltose and one fermenting slowly. They then performed an analysis of the transcriptome of these strains to identify transcription factors that may explain these differences between strains. In addition, they characterized the transcription binding sites profiles of each strain with an ATAC-seq procedure and identified the differences in chromatin accessibility between quickly fermenting and slowly fermenting strains.

The study is well led, and the manuscript clearly written. Exploring the bases of the phenotypic differences (here the switch of glucose to maltose) can be considered as a classical topic, but the authors presented here an interesting combination of cutting-edge technologies. This makes this study original in many aspects. However, I think that there is a general weakness in the lack of explanation of choices made. This could be said for the techniques, the fermentation stage, of the pair of genes chosen for performing analyses. Given the experience of the team with genetic analyses in yeast, why did the authors choose not to use a QTL analysis? What are the advantages of the actual strategy... this should be discussed. I also wonder why the choice of HAP5 and not HAP4 to be tested in combination with *cin5*. The "higher phenotypic differences obtained for hap5" are not obvious. The double deletant *hap4cin5* should have been tested too.

Minor comments

Page 15 Line 269: Differences in chromatin accessibility and transcription factor binding between *S. eubayanus* strains
The reasons of the choice of these two stage 24 and 72 h should be explained. I

Page 19 line 343 353 The choice of the combination of hap5 and cin5 is not obvious. Why not hap4 which deletion leads to clear increase in growth rate and a decrease in the lag phase ...

Figure 1B : the color codes used for strains has not been kept in figure 1A and 1B: this makes more difficult the reading

In figure 3B, I do not understand the differences between the sum of categories between differentially expressed genes and the number of up or down regulated genes

In figure 4 correct Accessibility

Molecular Genetics Laboratory
Chemistry and Biology Faculty
University of Santiago, Chile (Usach)
Santiago
Chile

October 5th, 2022

Dear Editors,

We want to thank the editor and the reviewers for the attention given to revise our manuscript "Natural Variation in Diauxic Shift between Patagonian *Saccharomyces eubayanus* Strains". We have now addressed the comments from all reviewers and made the suggested changes to improve our manuscript.

Below is a point-by-point response to the questions raised.

Yours sincerely,

Francisco A. Cubillos
Associate Researcher
Chemistry & Biology Faculty
Universidad de Santiago de Chile
+56227181084
Santiago, Chile

Reviewer #1

Molinet et al investigate strain differences in fermentation profiles of wild isolates of *S. eubayanus*. They characterize differential expression and chromatin accessibility in two high ferment strains and one low ferment strain. They identify groups of genes that are differentially expressed underlying the diauxic shift from glucose to maltose, a process that is important in fermentation. They identify four important transcription factors and create null mutants in relevant backgrounds, showing that *CIN5* is a repressor of diauxic shift in the low ferment strain. However the *CIN5* effect seems to be dependent on the carbon source and the genetic background, so while these results are really interesting from an evolutionary perspective, how this might be translational is unclear. Nevertheless, I thought overall this was a clear and comprehensive study.

R: First of all, we would like to thank the reviewer for the useful comments on our manuscript. We have now addressed most of the comments and performed the suggested changes.

A further elaboration of phenotypes in *S. cerevisiae* (and/or *S. pastorianus*) and how they compare with *S. eubayanus* both in terms of fermentation phenotypes, diauxic shift, and transcription factor mutants would provide helpful context. I'm assuming that diauxic shift from glucose to maltose is well described in *S. cerevisiae* (but perhaps not incorporating strain diversity?). What does this new work add to the field, and what can you say about how the key players have diverged or been conserved functionally?

R: Certainly, diauxic shift has been studied in greater detail in S. cerevisiae than in other species. However, in most cases the experiments are performed under laboratory conditions and not necessarily under fermentative conditions. The majority of these studies addressed the switch between glucose- galactose, and to a lower extent the glucose-maltose switch in laboratory strains.

Recently, Kevin Verstrepen's group reported the diauxic shift from glucose-galactose in 18 different S. cerevisiae strains (Perez-Samper et al., 2018). Indeed, their results indicate that the lag phase is dependent on the genetic context. However, they did not delve into the genetic differences that underlie these phenotypic differences.

Interestingly, in the same manuscript they analyzed the BY4742 yeast deletion collection, and found that null mutants of the HAP complex generate a lengthening in lag phase. These results are in contrast with those observed in the S. eubayanus QC18 strain, where hap4 and hap5 mutants decrease the lag phase, likely suggesting a different regulatory mechanism in this strain.

Therefore, we believe our study is the first of its kind to report how natural variation in Saccharomyces impacts the glucose-maltose and fermentative profile of different strains.

We have now incorporated this information in the discussion section and it reads as follows:

*'Previous reports demonstrated in the BY4742 haploid *S. cerevisiae* strain that null mutants of the HAP complex generate a lengthening in the duration of the lag phase (25). However, whether this phenotype is conserved across other isolates is unknown. Indeed, our results suggest that this pattern might not be conserved across isolates or even species in the *Saccharomyces* genus, and alternative regulatory mechanism might operate under different genetic backgrounds. While the HF knockout strains showed similar phenotypes to those described in *S. cerevisiae*, i.e. lag phase lengthening in HAP complex mutants (25), the QC18 strain showed a contrasting phenotype by significantly improving its growth rate in the glucose-maltose shift in *hap4Δ* and *hap5Δ* null mutants. Part of this alternative mechanisms could be upstream of the HAP complex regulation. In this sense, our analysis also highlighted the role of *Cin5p* as responsible for differences across strains, where *cin5□□* null mutants showed a shorter lag phase in HF strains. *Cin5p* is a basic leucine zipper TF of the *yAP-1* family, which participates in several stress conditions, including oxidative and osmotic stress (58).'*

The main conclusions of this study focus more on what's wrong with the low fermenting strain (QC18) than on what is better about the high fermenting strains, and it's unclear to me if the high fermenting strains are actually high fermenting, or just relative to the poor fermenting strain. In this respect, I think that the variation between the 2 high fermenting strains analyzed is an interesting path for further exploration for alleles that may be relevant for brewing strain improvement, since the authors showed that a similar fermentation phenotype may be regulated by different molecular mechanisms. This could use further discussion.

R: This is a very helpful observation. Indeed, we were scientifically intrigued on the genetic basis of the QC18 low performance, mostly because slow beer fermentations in non-cerevisiae strains is a main problem to obtain novel strains for the industry. Instead, we have now taken the reviewer's suggestion and we included new sentences in the results and discussion sections comparing the results obtained for the high fermentative strains.

These sentences now read:

'HF strains also differed in their expression patterns. Clusters III to VI showed different expression levels between both HF strains. For example, Cluster III contained genes related to carbohydrate transport upregulated in CBS12357 and downregulated in CL467.1 and QC18. On the other hand, Cluster IV contained genes related to response to stress and fungal-type cell wall organization downregulated in CBS12357 and upregulated in the other two strains (Fig. 2, Table S4B, S4C). These results suggest different regulatory mechanisms and molecular responses to achieve high fermentation levels.'

'HF strains exhibited differences between their chromatin accessibility patterns, represented in Clusters II and III (Fig. 4A). In Cluster II, we observed promoters showing higher accessibility in CBS12357, including genes related to maltose metabolism (MAL31, MAL32, IMA1), transporters (VBA5, HXT10) and ion homeostasis (FET4, ENB1). In cluster III, we identified promoter regions with higher accessibility in CL467.1, however these genes were unrelated to a particular metabolism function.'

I thought there were several points of disconnect about identifying important alleles for fermentation improvement, when really the only alleles discussed was the QC18 CIN5 allele and it was in a very specific context of a poor fermenting strain under non-fermenting conditions.

R: We agree on this, and we now included different genes either up- or down-regulated in the HF strains, which could represent potential important alleles in these backgrounds. These sentences were included in the results section as mentioned in the previous answer.

What is the architecture of MAL genes in *S. eubayanus* - how many MAL gene clusters are there? Is there copy number variation? Could certain alleles be non-functional in QC18?

*R: We have now included different antecedents and discussed our results including information about MAL genes in *S. eubayanus*.*

We recently sequenced the QC18 and CL467.1 strains, however, we are preparing a new manuscript with this information including QTL mapping, for this reason we rather prefer not to publish this information in the current manuscript. That being said, we did not find copy number variation or non-functional alleles in any of these backgrounds relative to the type strain. Therefore, we did not include this information in the current manuscript, and instead we incorporated the relevant information for the type strain in different sections across the manuscript.

Lines 251-263: Perhaps this section could be reworded and clarified to make it more clear why these transcription factors were selected, especially because this dictates the remaining experiments.

R: We have included this information in the results section, and now reads:

'To further identify transcriptional regulators underlying the fermentative differences between the three strains, we selected TFs based on four different criteria: i) significant binding differences predictions in DEGs between strains, ii) the presence of polymorphisms in the coding regions, iii) differences in expression levels across backgrounds under fermentation conditions and iv) literature supporting their role in diauxic shift and stress responses during fermentation. In this way, we identified four TFs: Hap4p, Hap5, Put3p and Cin5p.'

Lines 264-265: It would be useful to highlight here that all but 1 nonsynonymous difference are between the QC18 strain and the reference. Allelic differences do seem to be important (as described later in the hemizyosity tests). Is the QC18 CIN5 allele found in any other *S. eubayanus* strains?

R: We have now included this information, and it reads as follows:

'Between the CBS12357 and QC18 strains, the four chosen TFs harbor 5, 2, 0 and 2 non-synonymous SNPs for CIN5, HAP4, HAP5 and PUT3 (Table S4G), respectively.

However, none of the amino acid substitutions were identified as deleterious to protein function (Table S4H). These results suggest that differences in expression levels can likely be explained by differences in polymorphisms within the regulatory regions of the target genes. '

Line 426: glucose is not a polysaccharide

R: This has been corrected.

Reviewer #2

In this manuscript, Molinet et al. study the mechanisms explaining the different abilities of South American *S. eubayanus* strains to ferment and switch from glucose to maltose. Starting from the comparison of 19 strains, the authors retained two strains fermenting quickly maltose and one fermenting slowly. They then performed an analysis of the transcriptome of these strains to identify transcription factors that may explain these differences between strains. In addition, they characterized the transcription binding sites profiles of each strain with an ATAC-seq procedure and identified the differences in chromatin accessibility between quickly fermenting and slowly fermenting strains.

The study is well led, and the manuscript clearly written. Exploring the bases of the phenotypic differences (here the switch of glucose to maltose) can be considered as a classical topic, but the authors presented here an interesting combination of cutting-edge technologies. This makes this study original in many aspects. However, I think that there is a general weakness in the lack of explanation of choices made. This could be said for the techniques, the fermentation stage, of the pair of genes chosen for performing analyses.

R: We thank the reviewer for the constructive comments and suggestions. We have now incorporated several explanations for the choices made at the different stages of the manuscript. For example, we explained the choice of 24 hours for RNA-seq.

'To explore global gene expression patterns that could explain fermentation differences between HF and LF strains, we performed RNA-seq analysis on samples collected 24 h after the beginning of the fermentation. This time-point represents the inflection point when cells switch from glucose to maltose.'

In the same way, we explained ATAC-Seq time-points and the criteria to choose candidate Transcription Factors across the text (please see reply to Reviewer #1). For example:

'Samples for ATAC-seq were collected after 20 and 72 h of wort fermentation to evaluate the chromatin accessibility during the diauxic shift. These time-points represent the pre and post glucose-maltose switch since glucose is consumed in the first 24 h of the fermentation, while maltose consumption starts after 48 h.'

We expect that these explanations better clarify the reasoning behind our strategy. Most of these sentences were incorporated in the results sections.

Given the experience of the team with genetic analyses in yeast, why did the authors choose not to use a QTL analysis? What are the advantages of the actual strategy... this should be discussed.

R: Indeed, this was part of the discussion during the experimental design. We believe that both strategies are valuable and provide complementary results. While QTL mapping allows the identification of large and small effect variants, this approach is more laborious and does not necessarily provide insights into the interactions between variants and the regulatory mechanisms underlying a complex metabolic process, such as the diauxic shift. Therefore, we decided to identify how the genetic backgrounds differently regulated their metabolic switch between carbon sources, coupling two techniques that rapidly provide evidence and results on this sense (please also consider that this study was performed during the lockdowns in Chile). We are considering using QTL mapping for future approaches to this issue.

We have incorporated an additional sentence in the discussion section, and now reads:

'These results demonstrate the power of coupling RNA-seq with ATAC-seq to identify genetic variants responsible for phenotypic differences between different backgrounds. Alternative approaches, such as QTL mapping, can allow identifying large and small effect variants (57)}. However, this approach is more laborious and requires generating a large set of segregants, together with genotyping and phenotyping efforts. Instead, the analysis of the parental strains using two-cutting edge techniques, such as RNA-Seq and ATAC-Seq, rapidly identifies the genetic determinants and regulatory variants underlying the phenotypic variation between strains.'

I also wonder why the choice of HAP5 and not HAP4 to be tested in combination with cin5. The "higher phenotypic differences obtained for hap5" are not obvious. The double deletant hap4cin5 should have been tested too.

R: Unfortunately, for some reason we were unable to cure the cas9-plasmid from the hap4 mutant. Therefore, we couldn't generate double mutants to test their effect. Since this could be a concern for readers, we incorporated this information in the methods section.

Minor comments.

Page 15 Line 269: Differences in chromatin accessibility and transcription factor binding between *S. eubayanus* strains. The reasons of the choice of these two stage 24 and 72 h should be explained.

R: As mentioned before in a previous answer, this was incorporated in the main text

Page 19 line 343 353 The choice of the combination of hap5 and cin5 is not obvious. Why not hap4 which deletion leads to clear increase in growth rate and a decrease in the lag phase ...

R: As mentioned before in a previous answer, this was incorporated in the main text

Figure 1B : the color codes used for strains has not been kept in figure 1A and 1B: this makes more difficult the reading.

R: We have now matched the colour codes in both figures.

In figure 3B, I do not understand the differences between the sum of categories between differentially expressed genes and the number of up or down regulated genes

R: Figure 3B contains nine columns, where each columns represents a replica from the RNA-seq experiment for each strain. Then, clusters obtained by Hierarchical clustering were subjected to a gene ontology (GO) enrichment analysis. These results are depicted in Figure 3B.

To better clarify this sentence, we modified the figure legend and this now reads:

'(B) Hierarchical clustering of DEGs in the three strains. The heatmap was generated using the z-score of expression levels in each comparison. Each row represents a given gene and each column represents a replica from a different strain. Clusters are annotated at the right, together with their gene ontology (GO) category.'

In figure 4 correct Accessibility.

R: This has been corrected.

November 10, 2022

Dr. Francisco A Cubillos
Universidad de Santiago de Chile
Santiago
Chile

Re: mSystems00640-22R1 (Natural Variation in Diauxic Shift between Patagonian *Saccharomyces eubayanus* Strains)

Dear Dr. Francisco A Cubillos:

Thank you for submitting your manuscript to mSystems. We have completed our review and I am pleased to inform you that, in principle, we expect to accept it for publication in mSystems. However, acceptance will not be final until you have adequately addressed one additional reviewer comment (see below for the comment from Reviewer #1).

Preparing Revision Guidelines

Sincerely,

Benjamin Wolfe

Editor, mSystems

Journals Department
Reviewer comments:

Reviewer #1 (Comments for the Author):

The authors addressed my previous comments and I enjoyed the revised manuscript. The addition of methodological choices and context for the study helped improve my understanding and interest. My only very minor suggestion is that the authors include an idea of what fermentation kinetics look like for *S. cerevisiae* or *S. pastorianus* (like Fig 1B). Are the HF strains performing similarly?

Reviewer #2 (Comments for the Author):

The corrections made in this new version of the manuscript answers my questions/comments.

Molecular Genetics Laboratory
Chemistry and Biology Faculty
University of Santiago, Chile (Usach)
Santiago
Chile

November 11th, 2022

Dear Editors,

We want to thank the editor and the reviewers for the attention given to revise our manuscript "Natural Variation in Diauxic Shift between Patagonian *Saccharomyces eubayanus* Strains". We have now addressed the comment from reviewer #1 and made the suggested changes to improve our manuscript.

Below is a point-by-point response to the question raised.

Yours sincerely,

Francisco A. Cubillos
Associate Researcher
Chemistry & Biology Faculty
Universidad de Santiago de Chile
+56227181084
Santiago, Chile

Reviewer #1

The authors addressed my previous comments and I enjoyed the revised manuscript. The addition of methodological choices and context for the study helped improve my understanding and interest.

First of all, we would like to thank the reviewer for the useful comments on our manuscript. We have now addressed the comment and performed the suggested change.

My only very minor suggestion is that the authors include an idea of what fermentation kinetics look like for *S. cerevisiae* or *S. pastorianus* (like Fig 1B). Are the HF strains performing similarly?

*For a better understanding of the fermentation kinetics of *S. eubayanus* strains, we have included in Fig 1B the kinetics of the commercial strain *S. pastorianus* W34/70. Commercial strains have a better fermentative capacity than *S. eubayanus*, since *S. eubayanus* does not consume maltotriose, the second most abundant sugar in beer wort.*

Reviewer #2

The corrections made in this new version of the manuscript answers my questions/comments.

November 16, 2022

Dr. Francisco A Cubillos
Universidad de Santiago de Chile
Santiago
Chile

Re: mSystems00640-22R2 (Natural Variation in Diauxic Shift between Patagonian *Saccharomyces eubayanus* Strains)

Dear Dr. Francisco A Cubillos:

Your manuscript has been accepted, and I am forwarding it to the ASM Journals Department for publication. For your reference, ASM Journals' address is given below. Before it can be scheduled for publication, your manuscript will be checked by the mSystems production staff to make sure that all elements meet the technical requirements for publication. They will contact you if anything needs to be revised before copyediting and production can begin. Otherwise, you will be notified when your proofs are ready to be viewed.

Publication Fees:

If you would like to submit a potential Featured Image, please email a file and a short legend to msystems@asmusa.org. Please note that we can only consider images that (i) the authors created or own and (ii) have not been previously published. By submitting, you agree that the image can be used under the same terms as the published article. File requirements: square dimensions (4" x 4"), 300 dpi resolution, RGB colorspace, TIF file format.

We recognize that the video files can become quite large, and so to avoid quality loss ASM suggests sending the video file via <https://www.wetransfer.com/>. When you have a final version of the video and the still ready to share, please send it to mSystems staff at msystems@asmusa.org.

Sincerely,

Benjamin Wolfe
Editor, mSystems

Journals Department
Supplemental Material 4: Accept
Supplemental Material 1: Accept
Supplemental Material 9: Accept
Supplemental Material 7: Accept
Supplemental Material 3: Accept
Supplemental Material 8: Accept
Supplemental Material 5: Accept
Supplemental Material 6: Accept
Supplemental Material 8: Accept
Supplemental Material 2: Accept